# Effect of Fermented *Artemisia argyi* on Egg Quality, Nutrition, and Flavor by Gut Bacterial Mediation

**DOI:** 10.3390/ani13233678

**Published:** 2023-11-28

**Authors:** Min Zhou, Lingyan Zheng, Tuo Geng, Yunfan Wang, Mijun Peng, Fengyang Hu, Jing Zhao, Xuesong Wang

**Affiliations:** Chinese Academy of Inspection & Quarantine Greater Bay Area, Zhongshan 528437, China; zhoum@caiqgba.org.cn (M.Z.); zly@caiqgba.org.cn (L.Z.); gengt@caiqgba.org.cn (T.G.); wangyf@caiq.org.cn (Y.W.); pengmj@caiqgba.org.cn (M.P.); hufy@caiqgba.org.cn (F.H.)

**Keywords:** hen industry, palatability, laying performance, microbiomic analysis, feed attractant

## Abstract

**Simple Summary:**

The palatability of *Artemisia argyi* was ameliorated by probiotics fermentation. Dietary supplementation fermented *A*. *argyi* (AAF) enhanced the laying performance of hens and kept the egg quality as normal. Additionally, egg amino acids and egg fatty acids altered by diet the addition of AAF, which was the result of gut microbiota regulation. Our results certified that AAF could serve as feed attractant in the hen industry.

**Abstract:**

To improve the palatability of *Artemisia argyi*, fermented *A*. *argyi* (AAF) were prepared by *Lactobacillus plantarum* and *Saccharomyces cerevisiae*, which were used in the hen industry subsequently. Six hundred hens were randomly divided into three groups: control (A), dietary supplementation AAF at a low level (B), and dietary supplementation AAF at a high level (C). After feeding for four months, egg production, egg quality, egg nutrition, egg flavor, plasma biochemical parameters, intestinal histology, and microbiome of the gut contents were analyzed among the three tested groups. Interestingly, 5–6 percentage points elevation in the laying rates were observed in the AAF-supplemented groups in comparison to the control, accompanied with a 5 g increase in daily feed consumption. Since no alteration in egg/body weights was detected, laying performance enhancement was the main effect of dietary supplementation AAF. Meanwhile, the compositions of the egg amino acids and fatty acids changed as the feed inclusion AAF changed, e.g., His and linoleic acid decreased almost 0.1 and 0.5 g/100 g, respectively, while oleic acid increased almost 0.4 g/100 g. In addition, although no significant difference was detected (*p* > 0.05), the *β*-diversity of the gut microbiota decreased as the diet addition of AAF decreased, and probiotics (*Faecalibacterium*, *Prevotellaceae*, *Intestinimonas*, and *Lachnospiraceae*) were the dominant keystone species under AAF treatments. These probiotics were well associated with the egg nutrition component variations based on the correlation analysis, as the Sankey plot showed. Furthermore, the results of headspace-gas chromatography-ion mobility spectrometry manifested that the egg volatile components varied (e.g., the contents of acetone, 4-methyl-3-penten-2-one, 1-hydroxy-2-propanone, ethyl acetate, ethyl octanoate, ethanol, and 2-butanol in the B and C groups were higher than in the A group) and separated clearly as daily supplementation AAF, indicating AAF hugely contributed to the egg flavor variation. Due to no significant differences noticed between the B and C groups, dietary supplementation AAF at a relative low level was enough to serve as a feed attractant in the hen industry for real feeding.

## 1. Introduction

According to statistics, China’s domestic egg consumption was 30.24 million tons in 2020, ranking first in the world [1]. Although the output of livestock and poultry products in China has also ranked first in the world for 20 consecutive years, it is an arduous task to meet this huge demand. Therefore, many methods have been adopted to increase the number of eggs, such as using antibiotics. Due to the merits of growth promoting and anti-disease, antibiotics were popularly employed in the poultry industry a few years ago [2,3]. With the help of antibiotics, the production of eggs can fully meet the needs of consumers, and it is estimated that the output of eggs will reach 32.81 million tons by 2024, with an annual output value exceeding 300 billion CNY. Unfortunately, China prohibited antibiotics market circulations in 2020 due to the types of problems expanding as time progressed, e.g., gene mutation, chronic toxicity, allergic reactions, cancer causes, and digestive system disorder [4,5,6,7]. Consequently, feed additives instead of antibiotics are urgently needed, and medical plants have been widely applied based on the properties of efficiency and safety to poultry production [8,9], such as *Ampelopsis grossedentata* [10], *Eucommia ulmoides* [11], and *Flos lonicerae* [12].

*Artemisia argyi* is a traditional medicinal plant belonging to the Asteraceae family, which has been used worldwide for more than 3000 years [13,14]. Due to the numerous physiological functions (including antioxidation, antitumor, anti-inflammation, anticoagulation, antiosteoporosis, immune regulation, etc.), *A*. *argyi* has revealed high medicinal and economic values [15,16,17]. These functions are well associated with its natural activity components of flavonoids, organic acids, terpenoids, polysaccharides, and coumarin [18,19,20]. Recently, *A*. *argyi* has been attempted to use as a feed additive in animal production because of its efficacy and activity, which could mitigate the adverse situation of antibiotic abuse [21]. It has been pointed out that a reasonable utilization of *A*. *argyi* in animal production can improve feed quality, decrease feed cost, and reduce environmental pollution [22,23,24,25]. However, although *A*. *argyi* has great potential in animal production, its palatability is poor, which limits its application to some extent [25].

At present, it is considered that probiotic fermentation can effectively elevate the palatability of traditional Chinese medicine plants and increase their effective active components [26]. Fermentation is a simple, cheap, and effective method with the advantages of antinutritional factors reduction, cell wall structure destroying, and effective components releasing, such as phenolic compounds [27]. Meanwhile, living microorganisms can colonize in animal intestines, which displays a competitive advantage in space and can continuously inhibit the growth of pathogenic bacteria [28,29]. Moreover, after microbial fermentation, the flavor and taste of plants will alter and thus increase the appetites of the animals [30]. Therefore, fermentation is an ideal method to help medicine plants serve as feed additive and improves palatability.

In the present study, fermented *A*. *argyi* (AAF) was formulated by *Lactobacillus plantarum* and *Saccharomyces cerevisiae* and used in the hen industry. Egg production, egg quality, egg nutrition, egg flavor, plasma biochemical parameters, intestinal histology, and the gut microbiota of hens were assessed between control and dietary supplementation with AAF. We hypothesize that dietary supplementation AAF contributes to egg production elevation via the palatability of *A*. *argyi* changing. Our study will provide a theoretical foundation for AAF application in the hen industry.

## 2. Materials and Methods

### 2.1. Preparation of Fermented A. argyi (AAF)

The dry leaves of *A*. *argyi* were purchased from Hengxin Co., Ltd. (Zhangjiajie, China) and fermented by a mixed bacterial suspension. The mixture of microbial suspension consisted of *Lactobacillus plantarum* and *Saccharomyces cerevisiae*, with a concentration of 1.0 × 10^9^ and 0.2 × 10^9^ CFU/mL, respectively. During fermentation, the solid–liquid ratio was 1:1, and the fermentation temperature was set at 35 °C for 48 h. After fermentation, the obtained product was dried at 50 °C for 24 h by an electro-thermostatic blast oven. The obtained extract was used in the animal experiments without otherwise specified.

### 2.2. Chemical Composition of AAF

Since AAF was a fermented product, the endotoxin was detected by using the relative kit (Jiangsu KeyGen Biotech Co. Ltd., Nanjing, China). Meanwhile, the concentrations of two typical activity components (polyphenol and flavonoid) were detected. The detection of polyphenol was conducted following GB/T 8313-2018, while the detection of flavonoid was according to the method described by Fu et al. [31]. Moreover, the detection of hexachlorocyclohexane (HCH); dichlorodiphenyl trichloroethane (DDT); quintozene (PCNB); *Salmonella*; *Staphylococcus aureus*; and heavy metals (Pb, Cd, and Hg) in AAF was carried out following the standard protocols described by the National Health Commission (NHC), such as NHC (2003) [32], NHC (2003) [33], NHC (2016) [34], NHC (2016) [35], NHC (2016) [36], and NHC (2014) [37], respectively.

### 2.3. Animal Experiment and Treatment

The animal experiments were conducted at Baishi Poultry Farm (Zhongshan, China). About 5-month-old laying hens (Jing Fen 1, *n* = 600) were randomly divided into three groups: A group was the basal diet; B and C groups were the basal diet supplementation with 200 and 500 mg/kg AAF, respectively. The composition and nutrient levels of the basal diet are listed in Appendix A, which was purchased from Zhengda Kangdi Co., Ltd. (Shenzhen, China). During the feed period, the room temperature was set at 24 ± 1 °C with a cycle of 16 h light and 8 h dark. Meanwhile, the hens were allowed free access to water. The types of indices (including egg number, egg weight, and daily feed consumption) were recorded during the entire feeding period of four months.

### 2.4. Sample Collection

Blood samples were obtained aseptically from the wing veins of hens and then centrifuged for 10 min at a speed of 3800 rpm. All blood samples were stored at −80 °C for further biochemical parameters detection. In addition, two organs (liver and spleen) were collected and weighed individually. Three intestinal tissues (duodenum, ileum, and jejunum) were sampled and stored in 4% paraformaldehyde. The cecum contents were harvested and preserved at −80 °C for further microbiomic analysis.

### 2.5. Laying Performance and Egg Quality Assessment

The laying rate (total number of eggs/number of hens × 100%), average egg weight, broken egg rate (BER, number of broken eggs/total number of eggs × 100%), body weight (BW, including initial and final), and daily feed consumption (DFC, total amount of feed consumed per day/hens number), as well as feed conversion rate (FCR, total feed consumption/total egg weight), were evaluated when the feeding experiment finished. Meanwhile, the related indicators of egg quality were assessed. The egg shape index, as well as eggshell thickness, were measured by an electronic digital caliper (LR44, Guangzhou, China). The shell weight was weighed by an electronic scale, and the shell strength was monitored by an eggshell strength tester (ST120H, Shengtai Ltd., Jinan, China). Simultaneously, an egg analyzer (EA-01, ORKA, Ramat Hasharon, Israel) was used to determine the yolk color and Haugh units. Additionally, the yolk and albumen water contents were detected in an oven, and the percentage of yolk was also calculated as follows: yolk weight/egg weight × 100%.

### 2.6. Determination of the Amino Acids and Fatty Acids of Eggs

The amino acids and fatty acids of cooked eggs were detected following GB 5009.124 and GB 5009.168, respectively. Additionally, the yolk cholesterol content was also detected according to the method described earlier [11].

### 2.7. Hematological Analysis

Three immunity indices (immunoglobulin A (Ig A), immunoglobulin G (Ig G), and immunoglobulin M (Ig M)) and three inflammatory factors (tumor necrosis factor-α (TNF-α), interleukin-1 (IL-1), and interleukin-6 (IL-6)) were analyzed by using relative commercial kits (Beyotime Biotechnology Ltd., Guangzhou, China). Meanwhile, four lipid-related indices (total cholesterol (TC), triglycerides (TG), low-density lipoprotein cholesterol (LDL-C), and high-density lipoprotein cholesterol (HDL-C)); four liver function-related indices (alanine aminotransferase (ALT), aspartate aminotransferase (AST), albumin (ALB), and alkaline phosphatase (ALB)); three nitrogen metabolism-related indices (UREA, blood urea nitrogen (BUN), and uric acid (UA)); and two shell strength-related elements (Ca and P) were also determined at Rayto Biotechnology Ltd. (Guangzhou, China) by using the relative commercial kits.

### 2.8. Safety Inspection of Eggs

Based on the national food safety standards of China, the detection of microbial (*Salmonella*, aerobic plate count, molds, and coliforms); Aflatoxin B1; and heavy metals (Pb and Cd) in eggs used GB 4789.4-2016, GB 4789.2-2016, GB 4789.15-2016, GB 4789.3-2016, GB 5009.22-2016, and GB 5009.74-2014, respectively.

### 2.9. Headspace-Gas Chromatography-Ion Mobility Spectrometry (HS-GC-IMS) Analysis

For egg volatile components detection, the HS-GC-IMS method was used. The egg samples were conducted on a gas chromatography-ion mobility spectrometer (GC-IMS) based on an Agilent 490 micro gas chromatograph (GC) (FlavourSpec^®^, Gesellschaft für Analytische Sensorsysteme mbH, Dortmund, Germany) equipped with an autosampler and headspace sampling unit. Meanwhile, the principal component analysis (PCA) and partial least squares-discriminant analysis (OPLS-DA) were used to test the differences in egg volatile components among the three tested groups.

### 2.10. Histopathological Examination

Three intestinal tissues (duodenum, ileum, and jejunum) were dehydrated with graded alcohol and xylene first and then embedded in paraffin. After cooling down, the solid paraffin was obtained and cut into 4-μm sections and further stained with hematoxylin and eosin. The morphology of the selected intestinal tissues was observed by an inverted microscope (ECHO, Chicago, IL, USA). Villus height and crypt depths were measured by using an automatic image analyzer (ECHO, Chicago, IL, USA).

### 2.11. Microbiomic Analysis

The DNA samples of the harvested gut contents were extracted by using a DNA isolation kit (Qiagen NV, Hilden, Germany). A forward primer (341F 5′-CCTACGGGNGGCWGCAG-3′) and reverse primer (805R 5′-GACTACHVGGGTATCTAATCC-3′) were applied to amplify the specific regions of extracted DNA with the following thermocycling program in three steps: first, denaturation at 98 °C for 30 s; second, 35 cycles of 98 °C for 10 s, 54 °C for 30 s, and 72 °C for 45 s; and last, a final elongation at 72 °C for 10 min. After amplicons purification and quantification, the V3-V4 region of the amplicons was applied to 16S rRNA sequencing on an Illumina HiSeq 2500 platform (Illumina, San Diego, CA, USA). The obtained clean data were denoised with the divisive amplicon denoising algorithm (DADA2), and sequences with 100% similarity were clustered into amplicon sequence variants (ASVs). The reference ASV sequences were in the SILVA database. The Bray–Curtis distances, together with the principal coordinate analysis (PCoA), were used to evaluate the *β*-diversity among the three tested groups based on the permutation multivariate analysis of variance (PERMANOVA). Co-occurrence networks were constructed to identify potential keystone species, with the threshold of the Spearman’s correlations at *r* ≥ 0.7 and *p* < 0.05. Random forest testing was performed to infer the potential keystone species with ten-fold cross-validation. Groups B and C were treated as NOT in comparison to the A group. Multiple correlations between the keystone species and imputed functional profiles, as well as amino acids and fatty acids, were calculated based on Spearman’s correlations with the threshold of *r* ≥ 0.3 and *p* < 0.05 and visualized as the Sankey plot. All plots (PCoA, network, random forest, triangle diagram, and Sankey plot) were generated by the package EasyMicroPlot in RStudio software (version 2022.07.0).

### 2.12. Statistical Analyses

Data were expressed as the mean ± SD and analyzed by one-way analysis of variance (one-way ANOVA) method (SPSS 19.0). Tukey’s test was applied to collate the results of multiple comparisons, with *p* < 0.05 considered statistically significant (SPSS 19.0). WPS Excel (Kingsoft office, Beijing, China) and GraphPad Prism version 9.4.0 (GraphPad Software, San Diego, CA, USA) were used to draw the figures.

## 3. Results

### 3.1. Material Basis of AAF

Two major active compounds (polyphenol and flavonoid) in AAF were determined, and their contents were 5.10 mg/g and 8.22 mg/g (Appendix A), respectively. Since AAF was a fermented extract, the endotoxin content was therefore analyzed and was lower than 15 eu. Meanwhile, the pesticide residue (HCH, DDT, and PCNB) and pathogenic bacterium (*Salmonella* and *Staphylococcus aureus*) were not detected in AAF. Additionally, the heavy metals (Pb, Cd, and Hg) in AAF meet the national food safety standards of China (Appendix A). The obtained extract (AAF) was used in the animal experiments unless otherwise specified.

### 3.2. Egg Production of Hens

No mortality of the hens was observed during the whole experiment. The laying rates in the B and C groups were 93.44% and 92.23%, respectively, which were nearly 5 percentage points higher than that in the A group (87.55%), suggesting AAF could enhance the egg production of hens (Table 1). Simultaneously, the broken egg rates (BERs) in the B (0.03%) and C (0.17%) groups were apparently lower than that in the A group (0.40%), suggesting that the daily supplementation AAF contributed to egg quality elevation. Although no significant difference was detected (*p* > 0.05), above 5 g upregulation in the daily feed consumption (DFC) was noticed, implying AAF could stimulate the appetite of the hens, which could potentially function as a feed attractant in the hen industry. Since the laying rate and DFC both increased in the B and C groups, the feed conversion rate (FCR) in these two groups was at the same level as that in the A group. In addition, the egg weight, body weight (BW) of the hens (including initial BW and final BW), and body weight-related organ (including liver and spleen) coefficients revealed no significant differences among the three tested groups (*p* > 0.05), implying egg production enhancement was the main effect of dietary supplementation AAF rather than changes in their weights.

### 3.3. Safety of Eggs

The pathogenic bacterium (*Salmonella*) was not detected in the eggs, and other microorganisms (including aerobic plate count, molds, and coliforms) in the eggs were all lower than 100, 3, and 10 CFU/mL among the three tested groups (Appendix A), respectively. In addition, no aflatoxin (aflatoxin B1) was determined, and the typical heavy metals (Pb and Cd) were below 0.01 mg/kg, indicating dietary supplementation AAF could maintain egg safety in the hen industry.

### 3.4. Quality of Eggs

After ensuring the egg safety, the egg quality was analyzed. As shown in Table 2, the ten tested indices (including yolk color, egg shape index, shell strength, shell weight, shell thickness, egg albumen height, Haugh unit, percentage of yolk, yolk moisture content, and albumen moisture content) displayed no significant differences among the three tested groups (*p* > 0.05), manifesting that the diet addition of AAF could maintain the normal quality of eggs.

### 3.5. Cholesterol of Eggs, and Lipid-Related Indices of Blood

No alterations in the yolk cholesterol contents were detected among the three tested groups (Figure 1). However, variations in the lipid-related indices of blood were examined. Yolk cholesterol and two lipid-related indices (LDL-C and HDL-C) revealed no significant differences among the three tested groups (*p* > 0.05). The content of TC significantly decreased in the B group in comparison to the other two groups, whereas the content of TG in the C group significantly increased (*p* < 0.05). Meanwhile, except for the lipid-related indices, the contents of the immunoglobulin indices (Ig A, Ig G, and Ig M) in the C group were significantly different than with those in the other two groups (*p* < 0.05). Additionally, the contents of blood P and Ca significantly dropped in the B and C groups when compared to the A group (*p* < 0.05), which might be related to the shell strength of the eggs. The contents of the other blood indices’ involvement with liver function (AST, ALB, and ALP); nitrogen metabolism (UREA, BUN, and UA); and inflammation (TNF-α, IL-1, and IL-6) had almost no significant differences among the tested groups (*p* > 0.05), except the content of ALT decreasing in the AAF-added groups (B and C groups).

### 3.6. Amino Acids of Eggs

Generally, five amino acids (ASP, Glu, Lys, Ser, and Leu) were the dominant amino acids, with their contents higher than 1.00 g/100 g among the three tested groups (Table 3). No significant differences were detected among the tested amino acids (*p* > 0.05), except the content of His in the B and C groups significantly decreased in comparison to the A group (*p* < 0.05). Moreover, the proportion of delicious amino acids (DAA) in all tested groups was greater than the standard of 40% (>43%), suggesting the eggs were of high quality.

### 3.7. Fatty Acids of Eggs

Overall, five fatty acids (including palmitic acid (C16:0), palmitoleic acid (C16:1n7), stearic acid (C18:0), oleic acid (C18:1n9c), and linoleic acid (LA; C18:2n6c)) were predominant fatty acids in all the groups, with their contents higher than 1.00 g/100 g, accounting for 94.68–94.89% of the total fatty acids (Table 4). Meanwhile, significant variations in the topmost fatty acids (oleic acid and linoleic acid) were determined, and a divergent alteration tendency was observed between the two fatty acids, such as oleic acid significantly ascended in the B and C groups when compared to the A group (increase almost 0.4 g/100 g), while linoleic acid descended (decrease almost 0.5 g/100 g) (*p* < 0.05). Moreover, downregulations in the palmitoleic acid and α-linolenic acid (C18:3n3) contents were also detected by dietary supplementation AAF, especially in the B group.

### 3.8. Volatile Components of Eggs

Seventy-four volatile components were identified from the sampled eggs of the three tested groups (Appendix A). As shown in Figure 2a, the main constituents in the eggs were aldehyde, alcohols, ketones, esters, and acids, accompanied by little furan. Of these, the contents of some ketones; esters; and alcohols (including acetone, 4-methyl-3-penten-2-one, 1-hydroxy-2-propanone, ethyl acetate, ethyl octanoate, ethanol, 2-butanol, etc.) were apparently higher in the AAF-added groups (B and C groups) in comparison to the control (A group). In the meantime, the PCA and OPLS-DA analysis exhibited that the volatile components separated clearly among the three tested groups (Figure 2b), which proved dietary supplementation AAF benefited egg flavor change, especially at a relative low dosage (B group).

### 3.9. Intestinal Histology

Three intestinal tissues (duodenum, ileum, and jejunum) were selected for histological analysis (Appendix A). Unfortunately, the villus heights and crypt depths of the three tested intestinal tissues exhibited no significant differences among the three tested groups (*p* > 0.05). In addition, except that the villus height to crypt depth rate of the duodenum significantly increased in the AAF-added groups (B and C groups) in comparison to the control (A group) (*p* < 0.05), the villus height to crypt depth rates of the ileum and jejunum revealed no significant differences among the three tested groups (*p* > 0.05). These results showed that the diet addition of AAF exerted no impact on intestinal absorption.

### 3.10. Gut Microbiome

Although no significant difference was monitored (*p* > 0.05), the beta-diversity of the gut microbiota declined in the AAF-added groups (B and C groups) according to the PCoA analysis (Figure 3a). Regardless that the relative abundance of Bacilli in one or two samples from the AAF-added groups (B and C groups) dominated the major niche, the relative abundance of bacteria displayed no remarkable differences at the class level (Figure 3b). Meanwhile, the top 30 gut microbiota at the genus level were selected (Figure 3c). Of these, 8 probiotics (*Lactobacillus*, *Prevotellaceae*, *Faecalibacterium*, *Muribaculaceae*, *Ligilactobacillus*, *Lachnospiraceae*, *Ruminococcaceae*, and *Intestinimonas*) and 10 opportunistic pathogens (*Bacteroides*, *Streptococcus*, *Megasphaera*, *Desulfovibrio*, *Clostridia*, *Parasutterella*, *Parabacteroides*, *Olsenella*, *Shuttleworthia*, and *Oscillibacter*) were observed, and most of them decreased as dietary supplementation AAF, except *Lactobacillus*, which was used for preparing AAF. Moreover, eight keystone species were screened based on the random forest test (Figure 3d), including *Faecalibacterium* (V9), *Olsenella* (V27), *Prevotellaceae* (V12), F082_unclassfied (V19), *Rikenellaceae*_RC9 (V3), *Intestinimonas* (V32), *Lachnospiraceae*_unclassified (V13), and *Parasutterella* (V20). Among them, half of the keystone species were probiotics. Furthermore, 13 predicted functional profiles imputed by PICRUSt2 (glycerate-2-kinase (COG2379), propanediol utilization protein (COG4869), glutamate synthase domain 1 (COG0067), acetaldehyde dehydrogenase (COG4569), S-adenosylmethionine decarboxylase or arginine decarboxylase (COG1586), kynureninase (COG3844), L-2-hydroxyglutarate oxidase LhgO (COG0579), N-acetylmuramic acid 6-phosphate (MurNAc-6-P) etherase (COG2103), pectate lyase (COG3866), N-acyl-D-aspartate/D-glutamate deacylase (COG3653), fructose-bisphosphate aldolase class 1 (COG3588), pyruvoyl-dependent arginine decarboxylase (PvlArgDC) (COG1945), and polygalacturonase (COG5434)) were applied to construct correlations with the keystone species (Figure 3e). Seven keystone species (including four probiotics) and six predicted functional profiles revealed significant positive correlations that would positively mediate the composition of amino acids and fatty acids (Figure 3f).

## 4. Discussion

### 4.1. Egg Production Elevation Ascribed to AAF Could Serve as a Feed Attractant in the Hen Industry

Numerous researchers have reported that dietary supplementation medical plant ferment extract can elevate the egg production of hens, such as the diet addition of pine needle extract (increase 9.4%) [38], ginger extract (increase 11%) [39], buckwheat extract (increase 9.4%) [40], and *Schisandra chinensis* extract (increase 8.4%) [41]. In the present study, the effect of *A*. *argyi* ferment extract on the egg production of laying hens was investigated. Interestingly, 5 to 6 percentage point increases in the laying rates were observed by daily supplementation AAF in comparison to the control, indicating dietary supplementation *A*. *argyi* ferment extract benefited in egg production reinforcement, which was in agreement with the above-mentioned ferment extracts. Different from egg production elevation, no variations in the body weights of the hens were found among the three tested groups, suggesting egg production elevation was the main effect of dietary supplementation AAF rather than increasing their body weights. The reason for this phenomenon might be ascribed to the daily feed consumption (DFC) of the hens increasing as the dietary supplementation AAF also increased. Additionally, due to the laying and growth performances being well associated with intestinal absorption, the intestine tissue morphology was analyzed. Since a higher villus height, as well as a higher villus height to crypt depth ratio, contributed to nutrients absorption based on intestinal histomorphology [42,43], we therefore measured the villus heights and crypt depths of the duodenum, ileum, and jejunum. Unfortunately, no significant differences in the villus heights, crypt depths, and villus height to crypt depth rates of the three intestinal tissues were determined between the AAF-added groups and control group, indicating *A*. *argyi* ferment extract had no impact on improving the intestinal absorption. Consequently, the egg production elevation was the result of DFC increase, which was in line with the feed conversion rate (FCR) being balanced among the tested groups [11], implying *A*. *argyi* ferment extract has good potential for serving as a feed attractant in the hen industry.

### 4.2. A Relatively Low Dosage of AAF Is Ideal for Maintaining the Normal Quality of Eggs

Due to the egg production being enhanced by dietary supplementation AAF, a co-occurrence concern about whether egg production elevation affected the egg quality was noted. Therefore, the types of indices related to egg quality were detected and revealed no differences among the tested groups, suggesting feed containing AAF could not affect the egg quality overall. To our surprise, downregulation in the broken egg rates (BERs) was found in the AAF-added groups, especially in the lower dosage-added group (B group), which was accompanied with a higher laying rate in the same group. The previous literature shows that adequate dietary Ca and available phosphorus (P) are essential for sustaining egg quality during long-term egg production [44]. Hence, the contents of blood Ca and P were assessed and were significantly lower in the AAF-added groups than in the control. Previous evidence pointed out that the right amount of Ca and P contents could ensure the quality of eggs [45,46]. Thus, a slight decline in the plasma Ca and P contents had no impact on the egg quality in this study. In addition, although no significant differences were monitored in the egg weights among the tested groups, almost 1 g weight loss was recorded in the AAF-added groups. Considering this aspect, it could be clarified that lower plasma Ca and P contents, as well as lower egg weights, were the associated results of egg production rising, which could maintain the egg quality as normal. Apart from egg quality, nutrition was another important factor for the eggs [47], e.g., yolk cholesterol, egg amino acids, and egg fatty acids. Similar to egg quality, the contents of the yolk cholesterol, total amino acids, and total fatty acids displayed no significant differences among the tested groups, showing that dietary supplementation AAF exerted no detrimental effect on egg nutrition. Moreover, due to the presence of food microorganisms, aflatoxin B1, and heavy metals inducing egg production and egg quality decreases [48,49,50], these indices were also detected, and they all met the national food safety standards of China. Remarkably, among all of these tests, the lower AAF-added group (B group) exhibited a better effect on the egg production and body health of the hens, such as the TC content significantly descending in the B group, whereas the TG content significantly ascended in the C group (higher AAF-added group), which was related to hyperlipidemia amelioration [51,52]. This might be attributed to the high concentration of *A*. *argyi* (leaves, 300 mg/kg) causing in vivo injury of the animals (e.g., mice), while the lower concentration (150 mg/kg) had a positive effect on the growth performance [53]. A similar adverse effect of *A*. *argyi* ferment extract on hens also determined that a higher dosage of AAF stimulated the immune response of hens (contents of Ig A and Ig M upregulated, while the content of Ig G downregulated). Considering these results, we concluded that dietary supplementation AAF at a relatively low level contributed to egg production elevation without affecting the quality, nutrition, and safety of eggs.

### 4.3. Impact of the Fermented Artemisia argyi on the Gut Microbiota

Although there were no differences in the total contents of the egg amino acids and egg fatty acids among the three tested groups, the compositions of these two nutrition components changed. For instance, the contents of His (bitter amino acids), palmitoleic acid, α-linolenic acid, and linoleic acid in the eggs significantly decreased via feed supplementation AAF, whereas the content of oleic acid significantly increased. Numerous works have demonstrated that a diet addition of a plant extract could participate in amino acid and fatty acid metabolism in laying hens, so that the flavors of the eggs changed [11,54]. Therefore, although the nutritional level of the eggs was balanced between the AAF-added groups and control, alterations in the egg flavor happened unquestionably. Former research reported that plant extracts could regulate the taxonomic and functional compositions of the gut microbiota, which further participated in mediating amino acid and fatty acid metabolism [55]. In the present study, the diversity of the gut microbiota descended as the feed supplementation AAF decreased, accompanied by the probiotics and opportunistic pathogens decreasing, especially the probiotics. Numerous studies have reported that probiotics were closely involved with amino acid and fatty acid metabolism [56,57], which might bring about variations in nutrition. Interestingly, half of keystone species were probiotics in this work and were positively correlated with the variations in the amino acids and fatty acids. Hence, it was clear that dietary supplementation AAF-induced nutritional component alterations were mediated by the gut microbial variations entirely. Additionally, due to the types of amino acids and fatty acids that will produce volatile compounds during metabolic transformation [58], alterations in the volatile compounds could reflect changes in the egg flavor. Our previous study established the fingerprints of volatile compounds in eggs through the HS-GC-IMS and PLS-DA methods and certified that these volatile components were significantly altered by dietary supplementation plant extracts [11]. Thus, the volatile components of the eggs were evaluated. Based on the HS-GC-IMS results, it unambiguously figured out that the profiles and concentrations of the egg volatile components varied as the feed inclusion AAF varied, especially at the relative low dosage (B group). Moreover, the PCA and OPLS-DA analysis showed that the volatile components were completely separated among the three tested groups, indicating that dietary supplementation AAF hugely contributed to the egg flavor changes. Taking into account the above-mentioned points, we therefore affirmed that dietary supplementation AAF induced the composition of nutrition component changes and contributed to egg flavor alterations by mediating the gut microbiota.

## 5. Conclusions

In the present study, the egg production, egg quality, egg nutrition, and egg flavor were investigated by the dietary supplementation of *A*. *argyi* ferment extract in the hen industry. Firstly, the laying rate was elevated via daily supplementation AAF, which was the main effect of feed consumption increase, implying that *A*. *argyi* ferment extract is an excellent feed attractant in the hen industry. Secondly, the egg quality remained normal, and body health amelioration was observed as dietary supplementation AAF was at a relatively low dosage. Thirdly, variations in the egg amino acids and egg fatty acids were closely involved with the gut microbiota regulation role of AAF, which majorly contributed to the probiotic alterations. Lastly, variations in the nutrition components changed the flavors of the eggs. These results pointed out that a diet addition of *A*. *argyi* ferment extract at a relatively low dosage is enough to serve as a feed attractant and can be widely performed in the hen industry. The palatability of *Artemisia argyi* was ameliorated by probiotic fermentation and applied in the hen industry, which could enhance egg production, as well as maintain egg quality.

## Figures and Tables

**Figure 1 animals-13-03678-f001:**
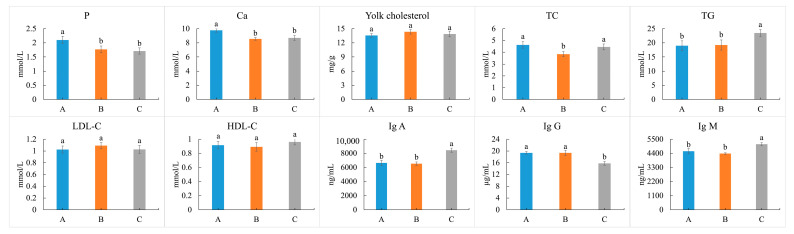
Effects of dietary supplementation AAF on the shell strength-related elements (Ca and P); lipid-related indices (yolk cholesterol, TC, TG, LDL-C, and HDL-C); and immunoglobulin indices (Ig A, Ig G, and Ig M). Different small letters manifested significant differences at the *p* < 0.05 level among the three tested groups.

**Figure 2 animals-13-03678-f002:**
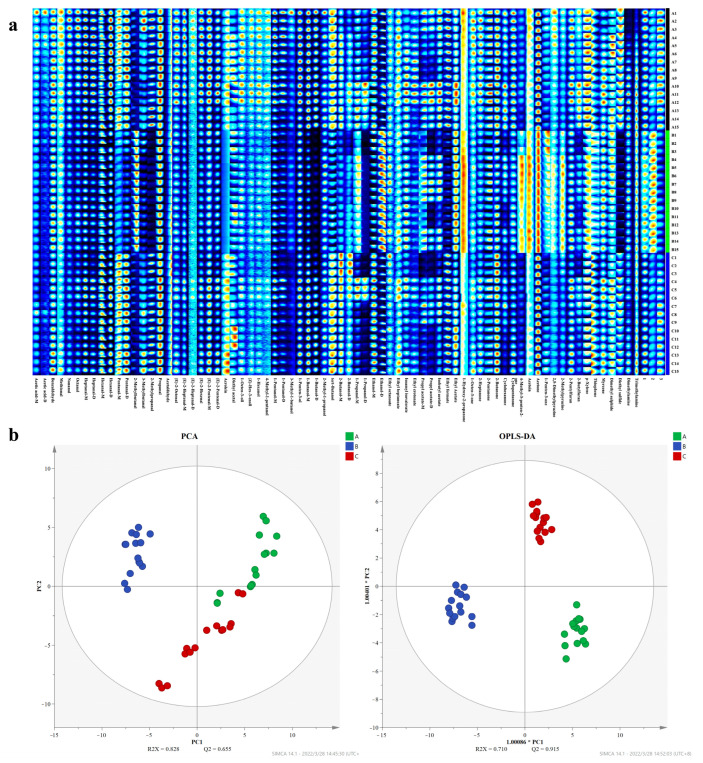
Effects of dietary supplementation AAF on the volatile components of eggs. (**a**) A gallery plot of the volatile components based on HS-GC-IMS of the three tested groups. (**b**) The PCA and OPLS-DA analysis of the volatile components of the three tested groups.

**Figure 3 animals-13-03678-f003:**
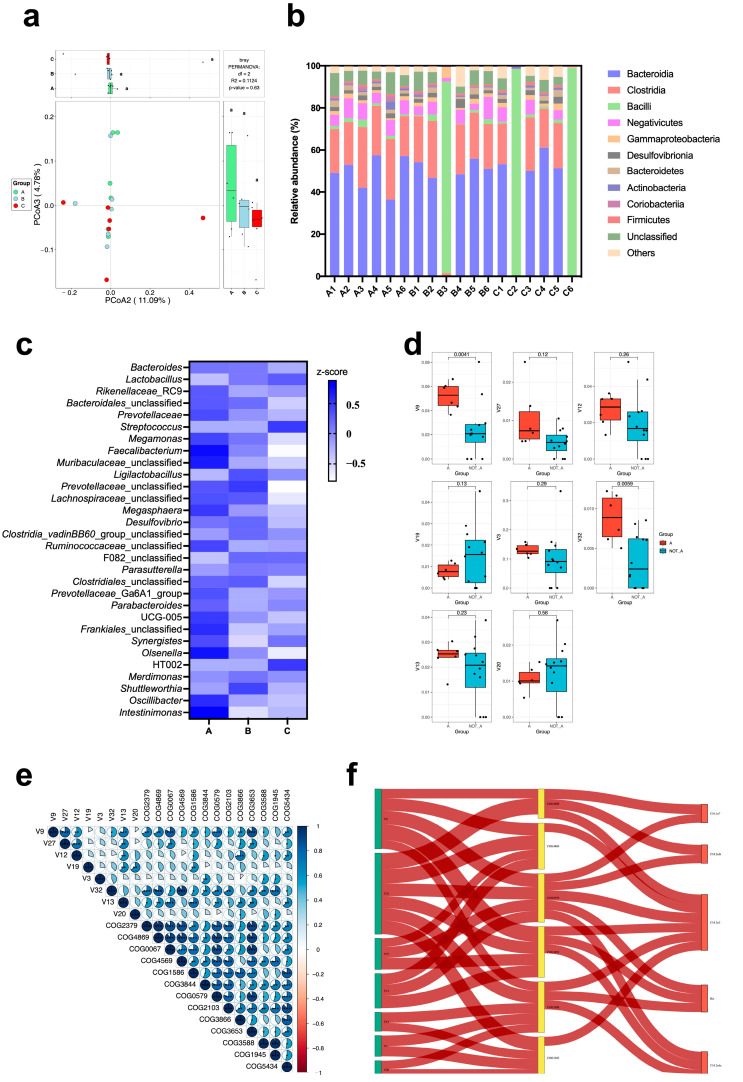
Microbial taxonomy, interaction, and keystone species of the gut microbiota among the three tested groups. (**a**) The principal coordinate analysis (PCoA) of the gut microbiota of the three tested groups. (**b**) The relative abundance of bacteria at the class level. (**c**) Heatmap of the top 30 gut microbiota at the genus level. (**d**) Keystone species based on random forest testing. (**e**) Correlations between the keystone species and predicted functional profiles imputed by PICRUSt2 (*, *p* < 0.05; **, *p* < 0.01; ***, *p* < 0.001). (**f**) Sankey plot shows the route of the gut microbiota participation in the egg nutrient alterations. Rectangles in green, yellow, and red colors represent the keystone species, imputed functional profiles of the gut microbiota, and amino acids or fatty acids in the eggs, respectively. Lines in red and blue mean the positive and negative relationships between categories.

**Table 1 animals-13-03678-t001:** Effects of dietary supplementation AAF on the laying and growth performances of the hens. Broken egg rate, BER; feed consumption, DFC; feed conversion rate, FCR; body weight, BW. Different small letters manifested significant differences at the *p* < 0.05 level among the three tested groups.

	Group	A	B	C
Laying performance	Laying rate (%)	87.55	93.44	92.23
	Egg weight (g)	62.43 ± 2.33 ^a^	61.63 ± 3.25 ^a^	61.35 ± 3.36 ^a^
	BER (%)	0.40	0.03	0.17
	DFC (g)	99.42 ± 6.65 ^a^	106.19 ± 9.69 ^a^	104.82 ± 8.46 ^a^
	FCR	1.83	1.85	1.86
	Initial BW (kg)	1.15 ± 0.14 ^a^	1.14 ± 0.20 ^a^	1.09 ± 0.15 ^a^
	Final BW (kg)	1.62 ± 0.11 ^a^	1.71 ± 0.14 ^a^	1.59 ± 0.14 ^a^
Organ coefficient	Liver (%)	2.53 ± 0.44 ^a^	2.43 ± 0.48 ^a^	2.52 ± 0.29 ^a^
	Spleen (%)	0.10 ± 0.01 ^a^	0.10 ± 0.01 ^a^	0.12 ± 0.02 ^a^

**Table 2 animals-13-03678-t002:** The egg quality-related indices among the three tested groups. Different small letters manifested significant differences at the *p* < 0.05 level among the three tested groups.

Items	A	B	C
Yolk color	6.60 ± 0.55 ^a^	6.60 ± 0.55 ^a^	6.80 ± 0.45 ^a^
Egg shape index	1.30 ± 0.02 ^a^	1.30 ± 0.02 ^a^	1.31 ± 0.04 ^a^
Shell strength (N)	38.79 ± 7.29 ^a^	36.51 ± 8.67 ^a^	37.35 ± 5.38 ^a^
Shell weight (g)	5.71 ± 0.33 ^a^	5.46 ± 0.33 ^a^	5.43 ± 0.36 ^a^
Shell thickness (mm)	0.32 ± 0.01 ^a^	0.31 ± 0.01 ^a^	0.31 ± 0. 01 ^a^
Egg albumen height (mm)	8.72 ± 0.87 ^a^	8.52± 0.63 ^a^	8.42 ± 0.93 ^a^
Haugh unit	92.50 ± 4.10 ^a^	92.34 ± 3.27 ^a^	88.16 ± 4.39 ^a^
Percentage of yolk (%)	25.86 ± 1.52 ^a^	26.95 ± 1.68 ^a^	25.73 ± 0.87 ^a^
Yolk moisture content (%)	48.81 ± 0.33 ^a^	48.90 ± 0.29 ^a^	48.16 ± 0.48 ^a^
Albumen moisture content (%)	88.25 ± 0.32 ^a^	88.32 ± 0.43 ^a^	87.93 ± 0.46 ^a^

**Table 3 animals-13-03678-t003:** The contents of egg amino acids as dietary supplementation AAF. Delicious amino acids, DAA; sweet amino acids, SAA; bitter amino acids, BAA; essential amino acids, EAA; total amino acids, TAA. Different small letters manifested significant differences at the *p* < 0.05 level among the three tested groups.

Item	Amino Acid (g/100 g)	A	B	C
DAA	Asp	1.38 ± 0.03 ^a^	1.36 ± 0.02 ^a^	1.39 ± 0.02 ^a^
	Glu	1.98 ± 0.03 ^a^	1.96 ± 0.03 ^a^	1.99 ± 0.02 ^a^
	Tyr	0.57 ± 0.01 ^a^	0.55 ± 0.02 ^a^	0.57 ± 0.03 ^a^
	Gly	0.47 ± 0.02 ^a^	0.45 ± 0.02 ^a^	0.47 ± 0.01 ^a^
	Phe	0.76 ± 0.01 ^a^	0.73 ± 0.03 ^a^	0.77 ± 0.02 ^a^
	Ala	0.77 ± 0.02 ^a^	0.75 ± 0.02 ^a^	0.79 ± 0.03 ^a^
SAA	Lys	1.09 ± 0.03 ^a^	1.05 ± 0.02 ^a^	1.07 ± 0.02 ^a^
	Pro	0.48 ± 0.02 ^a^	0.46 ± 0.02 ^a^	0.48 ± 0.01 ^a^
	Ser	1.03 ± 0.02 ^a^	1.02 ± 0.03 ^a^	1.04 ± 0.03 ^a^
	Thr	0.67 ± 0.01 ^a^	0.66 ± 0.02 ^a^	0.69 ± 0.04 ^a^
BAA	Val	0.86 ± 0.03 ^a^	0.83 ± 0.02 ^a^	0.87 ± 0.03 ^a^
	Leu	1.15 ± 0.03 ^a^	1.12 ± 0.04 ^a^	1.18 ± 0.04 ^a^
	Met	0.44 ± 0.01 ^a^	0.41 ± 0.02 ^a^	0.44 ± 0.01 ^a^
	Arg	0.88 ± 0.02 ^a^	0.88 ± 0.01 ^a^	0.88 ± 0.01 ^a^
	His	0.44 ± 0.01 ^a^	0.36 ± 0.01 ^b^	0.34 ± 0.01 ^b^
	Ile	0.69 ± 0.02 ^a^	0.68 ± 0.01 ^a^	0.70 ± 0.02 ^a^
EAA/TAA (%)		41.43	41.01	40.99
DAA/TAA (%)		43.41	43.58	43.91

**Table 4 animals-13-03678-t004:** The contents of egg fatty acids as dietary supplementation AAF. Different small letters manifested significant differences at the *p* < 0.05 level among the three tested groups.

Items		A	B	C
Fatty acids (g/100 g)	C14:0	0.11 ± 0.02 ^a^	0.12 ± 0.01 ^a^	0.12 ± 0.01 ^a^
	C14:1n5	0.03 ± 0.01 ^a^	0.03 ± 0.01 ^a^	0.03 ± 0.01 ^a^
	C15:0	0.02 ± 0.01 ^a^	0.02 ± 0.01 ^a^	0.03 ± 0.01 ^a^
	C16:0	9.03 ± 0.21 ^a^	9.01 ± 0.12 ^a^	9.02 ± 0.17 ^a^
	C16:1n7	1.31 ± 0.03 ^a^	1.21 ± 0.02 ^b^	1.26 ± 0.03 ^ab^
	C17:0	0.05 ± 0.01 ^a^	0.05 ± 0.02 ^a^	0.05 ± 0.01 ^a^
	C18:0	2.64 ± 0.11 ^a^	2.71 ± 0.11 ^a^	2.73 ± 0.13 ^a^
	C18:1n9t	0.03 ± 0.01 ^a^	0.03 ± 0.01 ^a^	0.03 ± 0.01 ^a^
	C18:1n9c	13.30 ± 0.12 ^b^	13.70 ± 0.14 ^a^	13.72 ± 0.13 ^a^
	C18:2n6t	0.01 ± 0.01 ^a^	0.003 ± 0.001 ^b^	0.004 ± 0.001 ^b^
	C18:2n6c	5.40 ± 0.21 ^a^	4.89 ± 0.16 ^b^	4.87 ± 0.14 ^b^
	C20:0	0.01 ± 0.01 ^a^	0.01 ± 0.01 ^a^	0.01 ± 0.01 ^a^
	C18:3n6	0.03 ± 0.01 ^a^	0.03 ± 0.01 ^a^	0.03 ± 0.01 ^a^
	C18:3n3	0.25 ± 0.02 ^a^	0.18 ± 0.02 ^b^	0.18 ± 0.01 ^b^
	C20:1n9	0.07 ± 0.01 ^a^	0.07 ± 0.01 ^a^	0.07 ± 0.01 ^a^
	C20:2n6	0.05 ± 0.01 ^a^	0.05 ± 0.01 ^a^	0.05 ± 0.02 ^a^
	C22:0	0.03 ± 0.01 ^a^	0.03 ± 0.01 ^a^	0.03 ± 0.01 ^a^
	C20:3n6	0.06 ± 0.01 ^a^	0.06 ± 0.02 ^a^	0.06 ± 0.01 ^a^
	C20:4n6	0.75 ± 0.08 ^a^	0.77 ± 0.04 ^a^	0.76 ± 0.06 ^a^
	C24:1n9	0.01 ± 0.01 ^a^	0.01 ± 0.01 ^a^	0.01 ± 0.01 ^a^
	C22:6n3	0.27 ± 0.02 ^a^	0.24 ± 0.03 ^a^	0.24 ± 0.03 ^a^
Total		33.46 ± 0.22 ^a^	33.22 ± 0.24 ^a^	33.30 ± 0.18 ^a^

## Data Availability

All datasets collected and analyzed in the study are available from the corresponding author upon reasonable request.

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
