# Peer review of "Effect of Fermented Artemisia argyi on Egg Quality, Nutrition, and Flavor by Gut Bacterial Mediation"

_animals, 2023, doi:10.3390/ani13233678_

Round 1

Reviewer 1 Report

Comments and Suggestions for Authors

1. Write a simple summary.

2. Lines 16-20; The Sentence is very long and confusing. 

3. The abstract lacks numerical data. 

4. Keywords should be different than phrases in the title.

5. Line 44; Popularly.

6. Lines 65, 82, 88& so on;  Italicize the taxonomic names of the organisms throughout the manuscript. 

7. Line 88; S. cerevisiae is not a bacterium. 

8. Line 100; How do they appear in the fermented AAF? Didn't the atuthors use sterilized material for fermentation, so only   Lactobacillus and Saccharomyces cerevisiae should be present in the AAF.  

9. Line 115; How do authors isolated plasma from the blood from veins? Elaborate the procedure. 

10. Line 120; it is better to highlight metagenomic sequencing rather than 16S rRNA sequencing. 

11. Line 132; How was it calculated? Please mention the formula. 

12. Lines 150-153; Check again for Salmonella, and rewrite for meaningful short sentences. 

13. Line 169; Change to Metagenomic analysis

14. Line 220; Table 1; Revise carefully, How are Organic coefficients and Egg weight statistically significant between groups? 

15. Lines 225-227; Revise for better meaning. If no Salmonella was detected, then how were lower than specified values?

16. Line 235; Significant is repetitive.

17.  Line 237, Table 2; Check the data of your tables very carefully. Most of the data is NOT statistically significant, correspondingly remove small letters from them.  

18. Line 239, 241, & so on; replace "indexes" with "Indices". 

Line 253, Figure 1; Replot the graphs, So much space between bars is not recommended. Increase the resolution and size of Axes titles.

19.  Lines 271-274; How much differences? Indicate values. 

20. Lines 279-300; Revise for clarity. 

21. Line 301; Correct the heading as Metagenomic analysis of the gut contents

22. Line 304; What was the abundance of Bacilli?

23.  Lines 307-316; Italicize the taxonomic names.

24. Lines 312-316; What was the frequency of the key stones species in the three tests. 

25. Line 401; Revise the heading, "Alterations in egg flavor induced by AAF are medicated by gut microbial" as " impact of the fermented Artemisia argyi on gut microbiota". 

26. Line 412; Amino acid and fatty acid metabolisms

27. Line 438; See if the phrase, "feedattractant" can be replaced with "appetizer".

Comments on the Quality of English Language

Many of the sentences are very long and confusing. It needs a careful revision by a native English speaker for proper grammar and content writing. 

Reviewer 2 Report

Comments and Suggestions for Authors

Dear authors, 

The manuscript was well-written and the content was informative and well-presented. I commend the authors for the comprehensive and systematic review of the topic. The manuscript will be a valuable contribution to this journal.

However, I’ve mentioned a few major comments that need to be addressed before the manuscript can be published: I also mentioned some minor corrections that need to be corrected in the comment section of the main manuscript file. Some of these are the following:

Line 9-15: Please write down a simple summary based on your current findings according to the journal guidelines.

Line 30-31: Please mention the dose of AAF which showed these beneficial effects.

Please add one line at the end of the abstract, which explains the basic output of this study and the future recommendations related to this study work as well.

Line 91-92: Please mention the drying temperature and time

Clarification of the specific research questions addressed in the study.

Line 191-192: Please mention the name of the statistical analysis software.

Line 253-255: Please explain a little bit more about the results from this figure.

Line 261: Please explain the term delicious amino acids. what kind of amino acids are included in this special category

Line 291-292: Please revise figure legends in detail

Line 295: Please clearly mention the specified part of the intestine, which part of the intestine, do you mean here

Line 331: Please revise the figure legends, as the picture is not clear.

Line 361: Please cite references that are in agreement or in a disagreement with your results in this discussion section

Please rewrite the conclusion part of this manuscript: To highlight the basic research gap that the authors try to cover in this study along with their future recommendations, based on their conclusion.

Line 460: Please set the entire list of references according to "Animals" MDPI journal instructions. The entire list of references is not according to the journal format.

Best wishes  

Round 2

Reviewer 1 Report

Comments and Suggestions for Authors

Although authors have made some changes to the original manuscript, many issues still need to be addressed.

1. Abstract still lacks numerical data, e.g., how much decrease/increase in the His, palmitoleic acid, and oleic acid contents? How much decrease in the Beta- diversity of the microbes? 

2. Line 174: replace "though" with "by". 

3. Line 458: space between feed and attractant. 

Reviewer 2 Report

Comments and Suggestions for Authors

Dear authors, 

The manuscript was well-written and the content was informative and well-presented. I commend the authors for the comprehensive and systematic review of the topic. The manuscript will be a valuable contribution to this journal in the current format after these corrections. 

best wishes, 
